# Manufacturing and Properties of Spherical Iron Particles from a by-Product of the Steel Industry

Andreas Walter [1,*], Gerd Witt [2], Sebastian Platt [2] and Stefan Kleszczynski [2,3]

1   Thyssenkrupp Steel Europe AG, Kaiser-Wilhelm-Straße 100, 47166 Duisburg, Germany
2   Institute for Product Engineering, University of Duisburg-Essen, Lotharstr. 1, 47057 Duisburg, Germany; gerd.witt@uni-due.de (G.W.); sebastian.platt@uni-due.de (S.P.); stefan.kleszczynski@uni-due.de (S.K.)
3   Center for Nanointegration Duisburg-Essen (CENIDE), Carl-Benz-Str. 199, 47057 Duisburg, Germany
*   Correspondence: andreas.walter3@thyssenkrupp.com

**Abstract:** In modern cold rolling mills in the steel industry, iron oxide powder is produced as a by-product when used pickling agents are recycled. Further processing of these iron oxide powders could enable the production of iron powder for various applications in powder metallurgy. For this purpose, a new process route with an eco-friendly hydrogen reduction treatment was developed. The process is able to manufacture a variety of iron particles through minor process adaptations. It was possible to manufacture spherical iron particles with high flowability. The flowability was measured by a Revolution Powder Analyzer, and an avalanche angle of 47.7° of the iron particles was determined. In addition, the bulk density measurements of the processed iron particles collective achieved values of 3.58 g/cm$^3$, and a spherical morphology could be observed by SEM analysis. The achieved properties of the iron particles show high potential for applications where high flowability is required, e.g., additive manufacturing, thermal spray and hot isostatic pressing. By adjusting the process conditions of the developed process, irregular iron particles could also be manufactured from the same iron oxide powder with a very high specific surface of 1640 cm$^2$/g and a low bulk density of 1.23 g/cm$^3$. Therefore, the property profile is suitable as a friction powder metallurgy material. In summary, the developed process in combination with the iron oxide powder from steel production offers a cost-efficient and sustainable alternative to conventional iron powders for additive manufacturing and friction applications.

**Keywords:** iron powder; hydrogen reduction process; iron oxide powder; by-product; hydrochloric acid pickling waste liquid recycling process

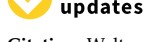



## 1. Introduction

Many different iron powder grades are available in the field of powder metallurgy due to different applications and manufacturing processes. Important indicators for evaluating the properties of metal powder grades are the particle morphology, the particle size, and the chemical composition. These essential particle properties depend on the manufacturing method, which is why various methods have been established for the production of iron powder. These methods are mechanical, chemical and electrochemical based [1]. The atomization techniques as a mechanical method and the solid-state reduction as a chemical method are most commonly used for a wide range of applications [1]. The atomization techniques use air, water, or gas as a carrier medium and offer the possibility to produce fine powder with a spherical-to-irregular particle morphology. Furthermore, the inner structure of the particle can reach a high density with this method. The second main process of producing iron powder is the reduction method, which is used when irregular and porous powder structures are required. Current conventional reduction methods are not suitable for producing spherical and dense iron powder. In addition, non-eco-friendly carbon-based reducing agents are typically used during the reduction from iron ores to

iron. Due to the committed European climate targets, which are described in the Green Deal [2] and Circular Economy Action Plan [3], the whole powder industry is working on sustainable processes intending to reduce greenhouse gases and save natural resources through a circular economy. Especially against the background of the increase in powder demand expected for the next decade, development activities with a focus on making processes sustainable are all the more important.

Several studies have investigated the use of scale as a source of iron oxide from the steel industry as an environmentally friendly raw material for the reduction processes [4–9]. Here, irregular and porous iron powders with Fe content below 99% are reported. The used scale is produced during the mechanical removal of the unwanted scale layer from the steel surface, for example before the hot or cold rolling process. For every ton of steel produced, about 65 kg of ferrous scale is accrued [10].

Further studies [11,12] are focusing on the use of $Fe_2O_3$ powder, which is a by-product from the recycling of conventional hydrochloric acid pickling liquid waste. In the conventional recycling process, the pickling waste liquid, which consists of ferric chloride and hydrochloric acid, is regenerated into reusable hydrochloric acid for the pickling line by the spray roasting process. During the spray roasting process, the pickling waste liquid is decomposed into hydrogen chloride gas and iron oxide powder at 673 K. The gas is absorbed and forms the regenerated acid and iron oxide powder, which is a by-product of this regeneration process [13,14]. Iron oxide powder from hydrochloric acid regeneration consists of $Fe_2O_3$ with an $Fe_2O_3$ content >99.0 wt.%. The mechanically removed scale usually consists of different iron oxides ($Fe_2O_3$, $Fe_3O_4$, and $FeO$), and the impurity content is higher compared to iron oxide from the acid regeneration. In addition to the difference in the chemical composition, the particle size of scale as an iron oxide source is much higher.

Walther et al. [11] and Danninger et al. [13] investigated the manufacturing of iron particles from iron oxide ($Fe_2O_3$) powder, which is a by-product of the recycling of conventional hydrochloric acid pickling liquid waste in the steel industry. They developed a new process on a laboratory scale for manufacturing spherical iron particles with a fine particle size distribution (PSD) and low inner particle porosity for the powder metallurgy (PM) industry [11].

Initially, the iron oxide powder was mixed into a suspension in a ball mill with the addition of binding and dispersing agents [11]. By atomizing the suspension with a spray drying system, spherical agglomerates of fine iron oxide particles were manufactured. Larger agglomerates of >30 μm were removed with a screening machine for the following reduction and sintering process steps. After sintering, the particles were further milled and subsequently sieved. The described process enabled the manufacturing of particles with a bulk density of 2.74 g/cm$^3$ and a PSD ranging from D10 = 13.2 μm to D90 = 35.6 μm.

For the spray drying process, a suspension typically consisting of iron oxide powder, distilled water and a carbon-based binder is prepared [15]. This suspension is specifically tailored to the process parameters of the spray dryer [15,16]. However, the carbon-based binder contaminates the iron oxide particles, leading to the carburization of the iron particles during the reduction and sintering process. In addition, unusable iron oxide particles that are sieved out before reduction and sintering due to their particle size cannot be spray-dried again because impurities in the form of binders are present on the iron oxide particles. Reusing the spray-dried iron oxide particles without removing the binder would change the composition of the entire suspension and have a significant impact on the process. A further disadvantage is the high initial investment cost of spray dryers [17]. For these mentioned reasons, an alternative cost-effective solution approach without contamination during the process is required.

Danninger et al. [12] chose a relatively low-effort process route without prior spray drying in their work to manufacture iron particles from iron oxide. With this process, they pursued the goal of producing sponge iron particles for conventional pressing and sintering applications and not spherical iron particles. The investigated process route includes a reduction step also with hydrogen at higher temperatures between 873 and 1573 K followed

by a grinding and screening process step. The investigations showed that at 1193 K the PSD has a dominant fraction D50 between 160–355 μm. The formed single particles were not significantly destroyed by the downstream grinding process. The bulk density was also measured after the 1193 K treatment, and values between 2.3 and 2.4 g/cm$^3$ were achieved. With a further investigation at a temperature of 1133 K during the reduction process with hydrogen and subsequent grinding, the PSD could be shifted to smaller particle sizes of <160 μm, but the bulk density of the particles decreased to 1.9 g/cm$^3$. Increasing the temperature above 1193 K led to a higher sintering effect between the single particles and consequently to a strong sinter cake, which cannot be used as particles [12].

The investigations show that the iron oxide from the recycling process of used pickling agents has a high potential to manufacture eco-friendly iron particles with different particle characteristics. However, there is a need for a cost-effective manufacturing route with the possibility to produce different iron powder properties, i.e., from spherical dense particles to irregular porous particles, and without a contamination of the particles due to the process route.

Therefore, the focus of this work is to develop a technologically reduced process route without the use of a spray drying process to produce spherical and dense iron particles from iron oxide powder obtained from hydrochloric acid regeneration. The process must also be able to produce irregular and porous particles. In addition, the chemical and physical properties of the iron particles produced by this process are investigated and compared with the properties of conventional water-atomized pure iron powder, which are used for many different technologies and applications, such as pressing and sintering, friction, welding, brazing, metallurgical cutting, additive manufacturing and hot isostatic pressing.

## 2. Materials and Methods

### 2.1. Materials

The iron oxide ($Fe_2O_3$) was produced during the regeneration process of acidic pickling waste in cold strip production at ThyssenKrupp Steel Europe. The $Fe_2O_3$ powder with a high purity of >99.5 wt.% was achieved by the removal of free acid and desiliconization process steps before roasting. The iron oxide powder consisted of fine primary particles that formed iron oxide agglomerates. Agglomeration was set through the thermal energy in the spray roasting process during hydrochloric acid regeneration.

### 2.2. Characterization

The laser diffraction method according to ISO 13320-1 was used to determine the particle size on the volume distribution of the investigated particles in this study [18]. Therefore, the laser diffraction spectrometer type Helos/BR (Sympatec GmbH, Clausthal-Zellerfeld, Germany) with a suspension cell was used. In the suspension cell, particles were dispersed in distilled water and transported through the measuring cell of the laser diffractometer. A helium–neon laser with a wavelength of 632.8 nm was used as laser source. For the measurement of the primary iron oxide particles, the iron oxide agglomerates were deagglomerated for 15 min with a lower milling intensity using an eccentric vibrating mill from (Siebtechnik GmbH, Mülheim an der Ruhr, Germany), which was equipped with 30 mm steel balls.

The specific surface area of the particles was measured according to DIN ISO 9277 by applying the Brunauer–Emmett–Teller (BET) method, which is based on a gas adsorption method [19]. The measurement was carried out with 1 g particle samples by using the Gemini 2360 instrument from Micromeritics (Micromeritics Instrument Corporation, Norcross, GA, USA). The bulk density of the particles was determined by using the funnel method according to ISO 3923 [20]. The flow properties of the iron particles were measured under static and dynamic conditions. For the static conditions, the Hall flow method according to DIN ISO 4490 was used to determine the flow rate of a particles quantity of 50 g through a funnel with a standardized outlet opening of 2.54 mm [21]. The "Revolution Powder Analyzer" (RPA) from Mercury Scientific (Mercury Scientific Inc., Newton, CT, USA) was

used to determine the avalanche angle of particles under dynamic conditions [22]. The measurements were carried out in a rotary drum at a rotation speed of 0.3 min$^{-1}$ with a particle volume of 100 cm$^3$. Furthermore, metallographic methods were used to characterize the outer and inner structures of the particle. For the detailed investigations of the outer particle structure, a scanning electron microscope (SEM) type MERLIN FE-REM from the company Zeiss (Carl Zeiss Microscopy GmbH, Oberkochen, Germany) was utilized. Hence, particle samples were distributed on an adhesive tape and the upper loose particles were removed with compressed air to investigate single particles. The inner particle structure was analyzed on previously polished specimens using optical microscopy. Additionally, the chemical composition of the iron particles was determined by three different methods. The carbon and sulfur contents were analyzed by infrared absorption methods with the type Bruker G5 Icarus (Bruker Corporation, Billerica, MA, USA) according to DIN EN ISO 15,350 [23]. The oxygen and nitrogen contents were determined by extraction methods with the type LECO TC500 (LECO Corporation, St. Joseph, MI, USA) according to DIN ISO 15351 [24]. The contents of the remaining elements were measured by inductively coupled plasma optical emission spectrometry (ICP-OES) with the type Spectro Atros (SPECTRO Analytical Instruments GmbH, Kleve, Germeny) according to DIN EN 10,351 [25]. The chemical composition of the iron oxide particles was determined by using ICP-OES after microwave digestion. With the weight loss due to the reduction, the degree of reduction was indirectly determined with the weight measurement after each reduction treatment of the iron oxide particles. To additionally confirm the complete reduction of the iron particles, X-ray diffraction (XRD) measurements with a diffractometer (equipped with Co-K$\alpha$ radiation (wavelength = 1.789 Å at room temperature) from Philips AG (Philips AG, Amsterdam, The Netherlands) were carried out to ensure complete reduction of the iron particles.

To evaluate and analyze the compressibility of the iron particles grades, disc-shaped pressed samples were manufactured with a uniaxial press. The die tool for manufacturing the pressed samples had a diameter of 18 mm. The pressing was carried out in the unlubricated condition at a pressing pressure of 200 N/mm$^2$. After pressing, the density of the samples was measured by using a geometric measurement with subsequent weight measurement. Afterwards, a heat treatment of the pressed samples was carried out at 1408 K for 1 h under a N$_2$/H$_2$ atmosphere (95%/5%). The sinter density was determined in the same way as the determination of the density of the pressed samples.

### 2.3. Equipments for Particle Manufacturing

For the deagglomeration of the iron oxide particles, an eccentric vibrating mill from Siebtechnik GmbH (Siebtechnik GmbH, Mülheim an der Ruhr, Germany) was used, which was equipped with 30 mm steel balls. The thermal agglomeration under air atmosphere was carried out in a batch furnace of the type KM410/13 from ThermConcept (ThermoConcept GmbH, Bremen, Germany). For particle sieving operations, a laboratory screening machine from the company Haver&Boecker, Type EML 200 (HAVER & BOECKER OHG, Oelde, Germany), was applied with mesh sizes of 63 μm and 250 μm. In addition, ultrasonic technology from Haver was used for the fine fraction < 63 μm on the screening machine. A universal laboratory tube furnace was used for the reduction and sintering processes. The furnace consisted of a quartz glass tube and could be heated up to 1423 K using an electric heater. The tube furnace allows for a continuous flow of reaction gas through the heated furnace chamber to react with the iron oxide particles. For the final homogenization process, a Retsch mortar grinder (Retsch GmbH, Haan, Germany) was used to remove the small amount of sintering between individual particles.

### 2.4. Manufacturing Method of Iron Particles from Iron Oxide Powder

Figure 1 illustrates the developed process under laboratory conditions. The process can be split into two main phases consisting of two or three sub-steps. The first is the preconditioning phase, consisting of sub-steps 1 to 3. In this phase, the physical properties

of the final iron particles, such as morphology, bulk density, internal porosity, and specific surface size, are significantly influenced. The second reduction and sintering phase, consisting of sub-steps 4 to 5, can be described as the transformation phase. In this phase, the preconditioned iron oxide particles are chemically reduced to iron particles with a reducing agent. In the subsequent writing, the developed process in this work for manufacturing iron particles is explained in detail.

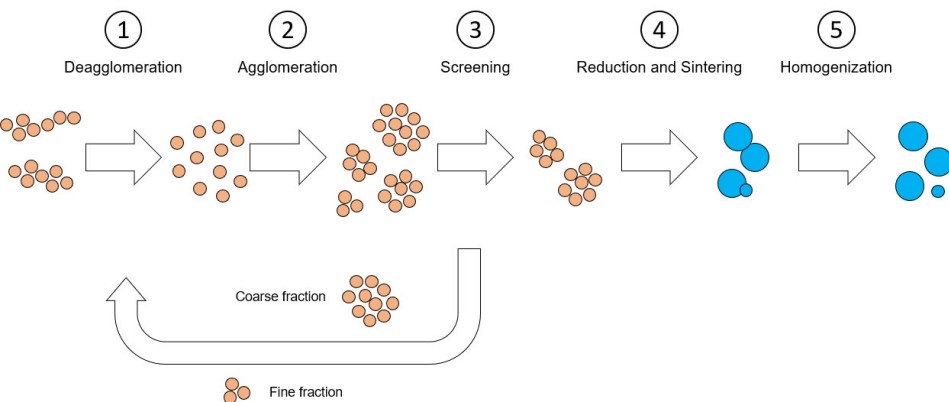

**Figure 1.** Manufacturing process of iron particles from iron oxide powder, from the recycling process of used pickling agent at a modern cold rolling mill.

The iron oxide particles produced during the recycling of pickling agents consisted of fine primary iron oxide particles that had agglomerated into coarse and stable iron oxide agglomerates with irregular morphology.

In the first step of the manufacturing route, the iron oxide agglomerates were mechanically deagglomerated with an eccentric vibrating mill down to the primary iron oxide particles with low intensity for a duration of 20 min. High milling intensities could cause primary oxide particles to partially re-agglomerate due to high temperatures.

Before the next step of heat treatment, the particles were loosely distributed on a flat stainless steel tray. Then, the deagglomerated primary particles in the tray were heated in a batch furnace under an air atmosphere until they reached 673 K within the core of the particles bed. The maximum bed height of the filled particles inside the tray was between 100 and 150 mm. After reaching 673 K, the tray with the particles was removed directly from the batch furnace and cooled down in an air atmosphere. The thermal energy causes the previously deagglomerated primary particles to rebuild iron oxide agglomerates during heat treatment. This can be explained by the increasing adhesion forces acting at the contact areas between individual primary particles [26]. The temperature-induced adhesion forces are attributed to different types of forces, which can also occur in combination. Other influencing factors besides temperature for the adhesion forces are particle size and morphology, as well as surface roughness, chemical composition, humidity, and distance between particles [27]. Already at 423 K, initial smaller portions of the fine iron oxide particles begin to agglomerate, but the proportion is very low. The agglomeration proportion steadily increases with a further temperature increase up to 623 K. A further increase in temperature above 623 K only causes a slight change in the particle size. Therefore, the temperature of 673 K is defined for the agglomeration step. The morphology of the newly agglomerated iron oxides is more regular and shows higher sphericity compared to the initial state.

The next step involved separating the coarse fraction from the agglomerated particles using a laboratory sieve machine with a 250 μm mesh size. This ensured that any particles that were too large could be removed prior to reduction and sintering. Furthermore, to investigate the influence of fine particles on particle properties during reduction and sintering, particles smaller than 63 μm were also separated for defined samples. During the fine screening, the screen housing was additionally equipped with ultrasonic technology to improve the fine screening process. After the screening step, the iron oxide agglomerates

were reduced with 99.999 wt.% pure hydrogen at 823 K and 1.5 h to iron particles, which can be described with the following two-step reaction for T < 843 K [28]:

$$3Fe_2O_3 + H_2 \rightarrow 2Fe_3O_4 + H_2O \tag{1}$$

$$Fe_3O_4 + 4H_2 \rightarrow 3Fe + 4H_2O \tag{2}$$

Typically, the reduction of $Fe_2O_3$ does not directly proceed to metallic iron particles. At temperatures below 843 K, reduction usually occurs stepwise, with the formation of $Fe_3O_4$ as an intermediate stage before the transformation to iron (Fe) continues [28], as illustrated in simplified form in Figure 2.

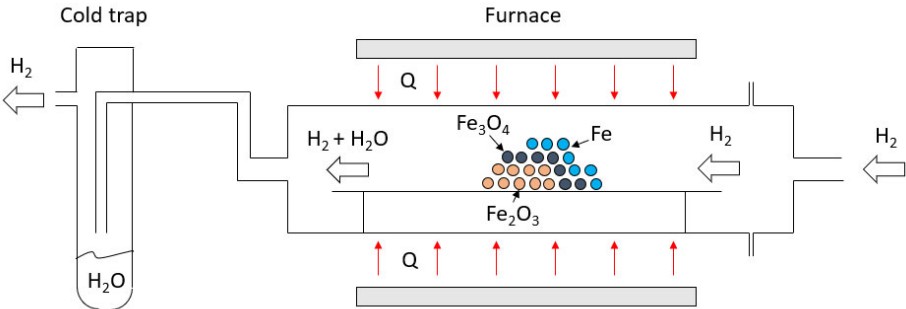

**Figure 2.** Schematic reduction process.

The hydrogen gas flowed through the tube furnace at a flow rate of 2 m$^3$/h, allowing the water vapor generated during the chemical reaction with the iron oxide to flow together with the hydrogen gas. A superstoichiometric amount of hydrogen gas is added, meaning that the amount of hydrogen is greater than what is required for the complete reduction of the iron oxide particles to iron particles. The outflowing gas was separated into hydrogen gas and fluid water through a cold trap with a condensate drain. After the reduction of the iron oxide particles, the temperature was increased to values between 973 and 1073 K for 1 h under an argon gas atmosphere to start the sintering step of the manufacturing, as illustrated in Figure 3.

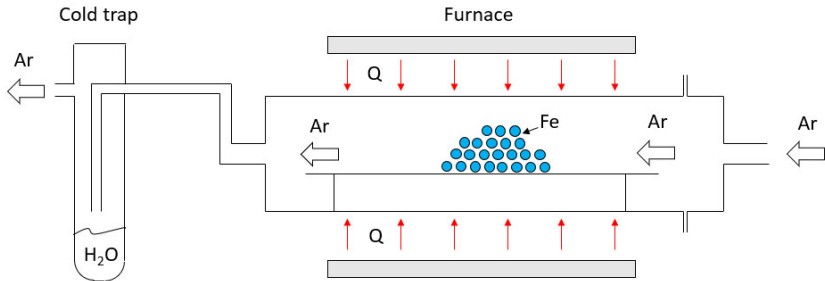

**Figure 3.** Schematic sintering process.

Increasing the temperature reduces the porosity of the reduced iron particles, which leads to a reduction in the specific surface size and shrinkage of the iron powder [11]. After cooling to room temperature, the iron particles were homogenized in a final process step in a low-intensity milling process for 10 min to remove the small amount of sintering between the single particles. Finally, the coarse fraction over 100 μm was separated from the particles with a screening machine. The following Table 1 summarizes the manufacturing route with the relevant process conditions.

The manufacturing process described above, along with the process conditions outlined in Table 1, was used to manufacture various samples of iron particles (HR1-HR6) by varying the process parameters as described in Table 2.

**Table 1.** Overview of process conditions for the manufacturing route.

| Preconditioning Phase | | | |
|---|---|---|---|
| | **Deaggomleration (1)** | **Agglomeration (2)** | **Screening (3)** |
| **Equipment** | Eccentric Vibrating Mill | Batch Furnace | Sieve Shaker |
| **Process parameters** | • Rotation speed 1500 1/min<br>• Duration 20 min<br>• 30 mm steel balls<br>• Steel tube<br>• Air atmopsher<br>• Dry condition | • Until core temperature of particles bed 673 K<br>• Air atmosphere | • Fine screening with mesh size of 63 μm equipped with ultrasonic technology<br>• Coarse screening with mesh size of 250 μm |
| Transformation phase | | | |
| | **Reduction (4)** | **Sintering (5)** | **Homogenization (6)** |
| **Equipment** | Tube furnace | Tube furnace | Mortar grinder |
| **Process parameters** | • Temperature 823 K<br>• Duration 1.5 h for 150 g particles batch<br>• Hydrogen atmosphere<br>• Gas flow rate 2 $m^3$/h | • Temperature 973–1073 K<br>• Duration 1 h<br>• Argon atmosphere | • Pistill pressure 80 N<br>• Rotation speed 70 1/min<br>• Duration 10 min<br>• Dry condition<br>• Steel milling set |

**Table 2.** Variation of process parameters for the manufacturing of iron particle samples.

| Sample | Variation of Process Parameters | | | | | |
|---|---|---|---|---|---|---|
| | **Milling** | **Agglomeration** | **Fine Screening (<63 μm)** | **Coarse Screening (>250 μm)** | **Reduction Temperature** | **Sintering Temperature** |
| **HR1** | Yes | Yes | No | Yes | | 973 K |
| **HR2** | Yes | Yes | Yes | Yes | | 973 K |
| **HR3** | Yes | Yes | No | Yes | 823 K | 1073 K |
| **HR4** | Yes | Yes | Yes | Yes | | 1073 K |
| **HR5** | No | No | No | No | | 973 K |
| **HR6** | No | No | No | No | | 1073 K |

For HR1 to HR4 iron particles, all sub-process steps (milling, agglomeration, sieving, reduction, and sintering) were carried out and two process parameters at the screening and sintering process step were varied. Furthermore, the screening (63 μm and 250 μm) of agglomerated iron oxide particles was varied. In addition, the subsequent reduction temperature was 823 K and the sintering temperature was varied between 973 K and 1073 K. For the manufacturing of HR5 and HR 6 iron particle samples, only the reduction and sintering process steps were carried out directly with iron oxide powder from the conventional recycling process with a variation of the sinter temperature between 973 K and 1073 K. All iron particle samples from HR1 to HR6 were homogenized at the end through a milling process.

## 3. Results and Discussion

### 3.1. Iron Oxide Powder from the Regeneration Process of Acid Pickling Waste Liquid

Figure 4 shows the surface structure of an iron oxide agglomerate from the regeneration process of acid pickling waste liquid consisting of fine primary particles. The primary particles have a size of <1 μm.

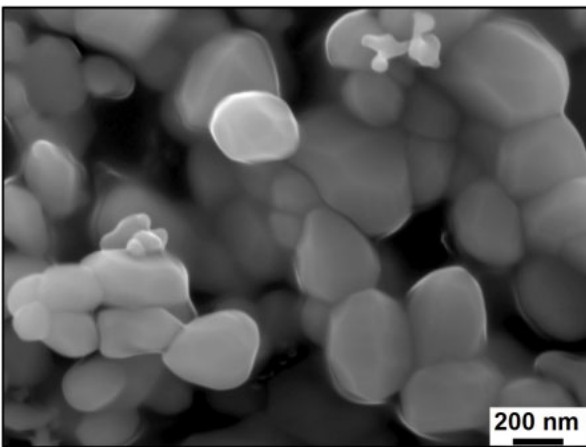

**Figure 4.** SEM analysis of iron oxide agglomerates.

The iron oxide agglomerate remains intact without the high deagglomeration effect during usually wet and dry handling processes due to sufficient mechanical stability. Therefore, the measurement of the PSD by the laser diffraction method was possible in the wet process. The particle size measurement of the iron oxide powder as an agglomerate showed a D10 value of 2.92 µm and a D90 value of 98.86 µm. The particle size determination of the deagglomerated primary particles resulted in a D10 value of 0.31 µm and a D90 value of 0.90 µm, which agree well with the SEM analysis in Figure 2.

*3.2. Influence of Process Parameters on the Bulk Density*

Figure 5 shows that the process conditions have a significant influence on the bulk density.

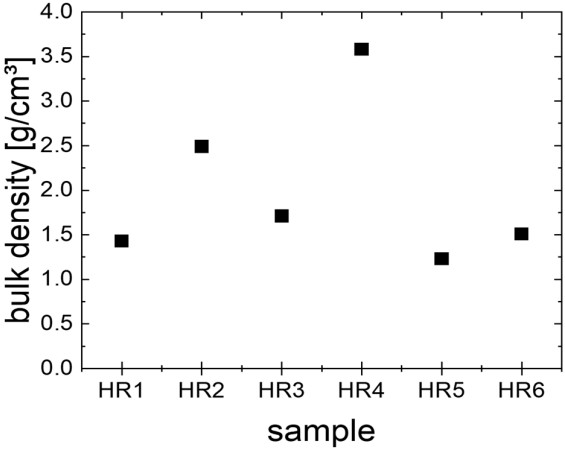

**Figure 5.** Bulk density of the manufactured particles.

The minimum bulk density was achieved for HR5 with 1.23 g/cm$^3$, and the maximum bulk density was achieved for HR4 with 3.58 g/cm$^3$. The influence of the sintering temperature can be seen in the higher bulk densities at higher temperatures. The increased bulk density was caused by the fine fraction of the iron oxide particles prior to the sintering. The particle samples HR5 and HR6 showed an irregular morphology compared to the samples from HR1 to HR4. Therefore, it was not considered for further investigation. For further investigations, the particles types HR1 and HR4 were selected.

### 3.3. Microscopic Characteristics of Iron Particles

Figure 6 shows the SEM images of the surface structure of the HR1 and HR4 iron particles. Shown for comparison, in Figure 6, is an SEM image of conventional water-atomized (WA) iron powder.

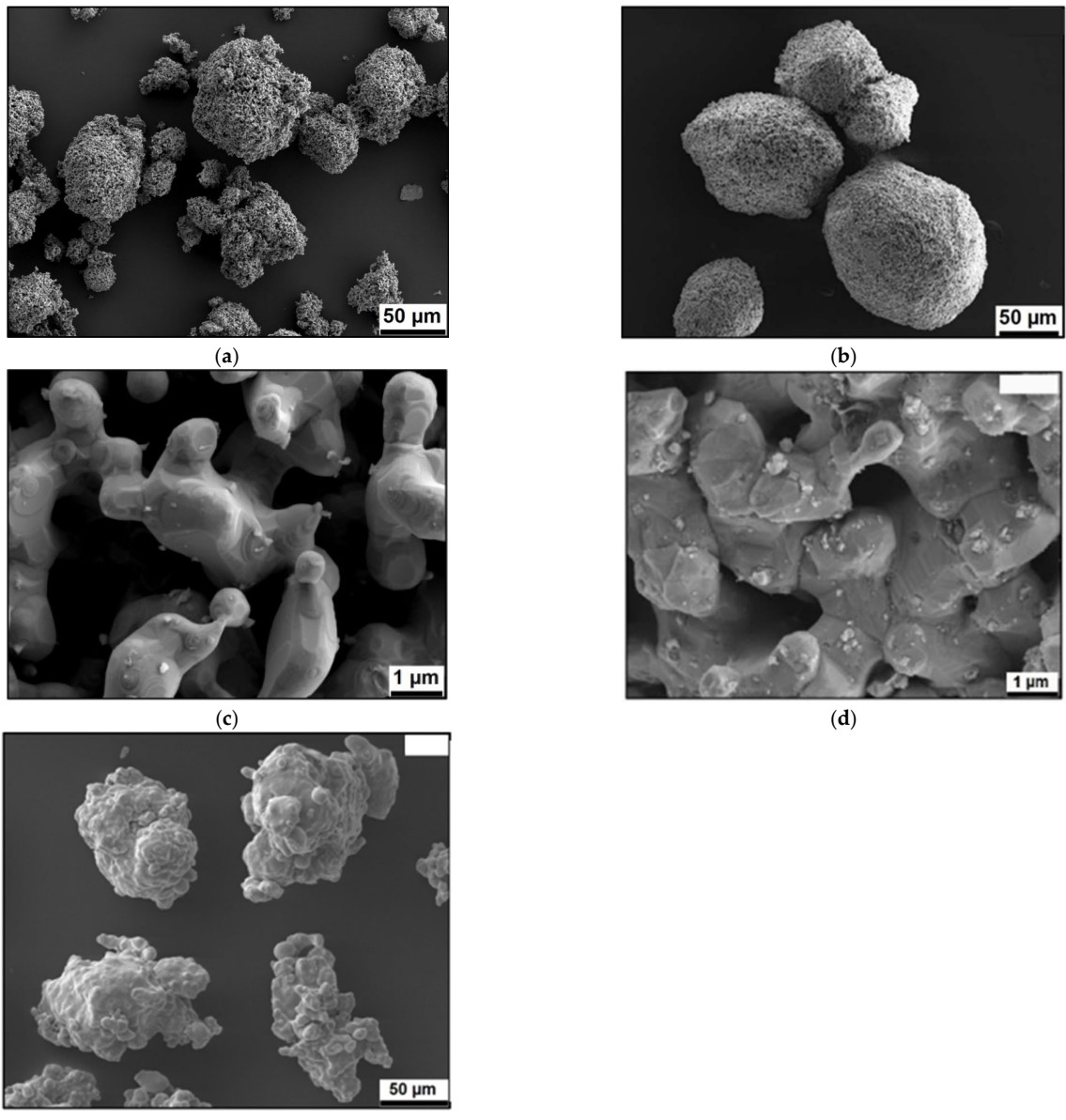

**Figure 6.** SEM analysis of HR1 (**a**,**c**), HR4 (**b**,**d**), and WA (**e**) iron particles with different resolutions.

The particle morphology between the HR1 and HR4 iron particles is different. HR4 achieved a regular particle morphology and HR1 showed an irregular particle morphology. Besides the differences in particle morphology, a higher open-pored surface structure can be observed in the HR1 iron particles compared to HR4 (Figure 4). Although an open-pored particle structure was observed for the HR4 iron particles, the size and number of open pores are smaller. In the case of the WA iron particles, a more dense surface structure could be observed but the morphology is more irregular compared to the HR grades. An exact interpretation of the inner structure of the HR1 and HR4 iron particles was possible with the SEM images of the prepared metallographic specimens, which can be seen in Figure 7.

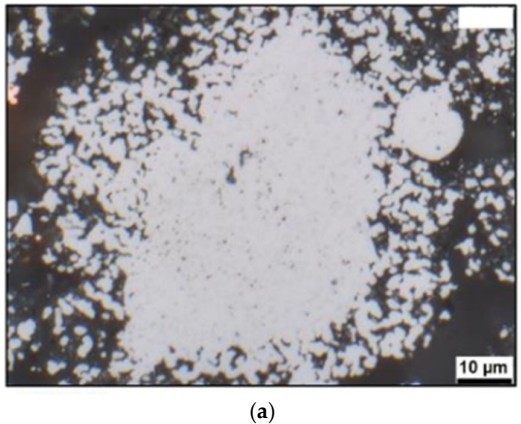 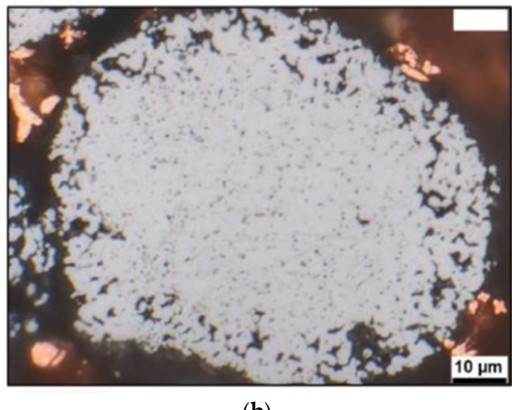

(**a**) (**b**)

**Figure 7.** Comparison of the micrographs of HR1 (**a**) and HR4 (**b**) iron particles.

Figure 7 shows an open-pored structure in the edge area of both iron particle grades. In both cases, the inner structure of the particles can be divided into two areas: the core area with higher density, and the edge area with higher porosity. The edge area of HR1 shows a significantly larger area, which leads to a higher inner porosity compared to the HR4 iron particles. The reason for the different particle properties is mainly due to the additional fine screening and the sintering temperature in the manufacturing process, whereby the fine screening has an increased influence on a particle's density. Without the fine screening step, loose primary particles or less agglomerated primary particles remain between larger compact agglomerates. After the complete reduction, the compact agglomerates shrink to dense particles due to the sintering effect at higher temperatures. Through this, the inner porosity of the particles is reduced. During the sintering process, the loose primary particles and small agglomerates are sintered on the surface of the larger compact agglomerates. After the complete reduction, and during the sintering process, the inner porosity of the compact agglomerates is reduced and the agglomerates shrink to compact particles with low porosity. At the same time, the loose primary particles and small agglomerates attach to the surface of the larger compact particles so that an edge area with higher porosity is built up. Nor can the porosity be significantly reduced by higher temperatures, as the gaps are larger.

### 3.4. Chemical Composition

Table 2 shows the results from the determination of the chemical composition of the HR1, HR4, and WA samples using infrared absorption and hot extraction methods, as well as ICP-OES. The pressing properties of iron particles are hardly influenced by the minor sulfur, oxygen and carbon content [29,30]. Therefore, the elements are listed in Table 3 for comparison of the contents. In addition to manganese, other elements were also determined by ICP-OES (e.g., Al, As, B, Co, Cr, Cu, Mg, Mo, Nb, Ni, P, Pb, Si, Sn, Ti, V, Zn), but their contents were very low and therefore negligible.

**Table 3.** Chemical compositions of iron particles samples.

|  | HR1 | HR4 | WA | Methods of Measurement |
|---|---|---|---|---|
| **C [wt.%]** | 0.006 | 0.0008 | 0.001 | Infrared absorption method |
| **S [wt.%]** | 0.007 | 0.004 | 0.006 | Infrared absorption method |
| **O [wt.%]** | 0.43 | 0.11 | 0.17 | Extraction method |
| **Mn [wt.%]** | 0.26 | 0.24 | 0.19 | ICP-OES method |

The Mn content of all iron particles are comparable. The oxygen and carbon content of HR4 are slightly lower compared to HR1 due to the longer heat treatment at higher temperature. In summary, the differences in chemical composition between the iron particles are small.

### 3.5. Physical Properties

Figure 8 shows the measured particle sizes according to their volume-weighted distribution.

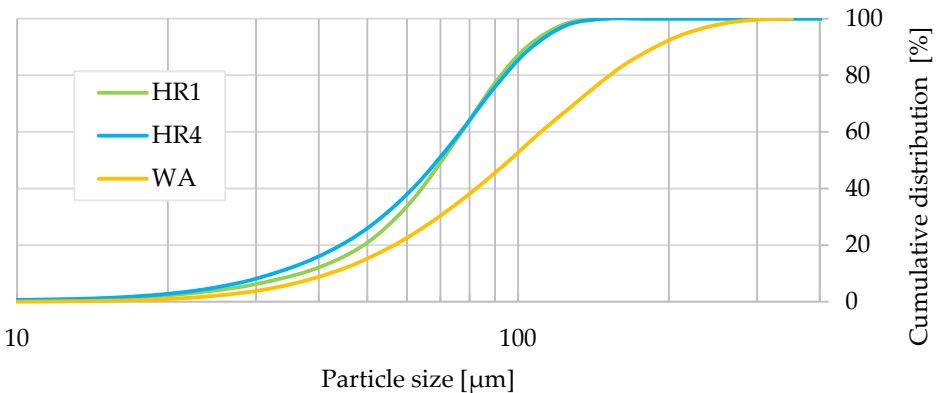

**Figure 8.** Particle size distributions (PSD) of HR1, HR4, and WA.

For better comparability of the PSD, the characteristic values are summarized in Table 4.

**Table 4.** Particle size distribution (PSD) D10, D50, and D90 of HR1, HR4, and WA.

| Samples | D10 [μm] | D50 [μm] | D90 [μm] | ΔD90/10 = D90–D10 [μm] |
|---------|----------|----------|----------|------------------------|
| HR1 | 36.75 | 70.62 | 104.33 | 67.58 |
| HR4 | 32.46 | 69.20 | 107.35 | 74.89 |
| WA | 42.02 | 96.10 | 188.01 | 145.99 |

The PSD of the HR1 and HR4 iron particles can be considered comparable to a first approximation. The distribution curve of HR4 is slightly wider compared to that of HR1. On the other hand, the WA iron particles has larger particles compared to both HR iron particles. The D90 value of the WA iron powder of 188.01 μm is significantly higher than the D90 value of the HR1 iron particles of 104.33 μm. The difference between D90 and D10 with 145.99 μm is twice as high compared to the HR iron particles.

Figure 9 shows the results from the determination of the bulk density using the funnel method according to ISO 3923 [20] and the flow rate using the Hall flow measurement method according to DIN ISO 4490 [21] of the considered iron particles in this study.

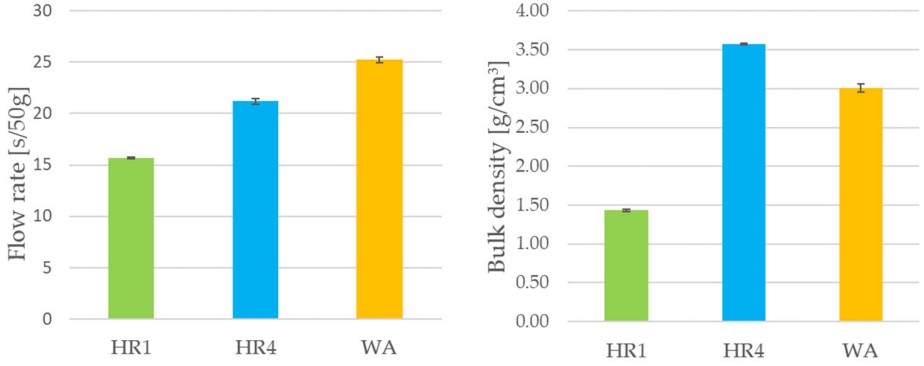

**Figure 9.** Bulk density and flow rate of the HR1, HR4, and WA iron particles.

The bulk density of the HR4 particles (3.58 g/cm$^3$) is higher than that of the HR1 particles (1.43 g/cm$^3$) and depends on the gaps between nearby particles and porosity of the particles. A more irregular particle morphology has the effect of reducing the bulk density, as larger gaps are usually created due to the irregular morphology. In addition, the

bulk density is reduced with increasing particle porosity. The HR4 iron particles has a more regular morphology and reduces internal porosity compared to the HR1 iron particles. This explains the large difference in bulk density between both HR particles. In addition to morphology and particle porosity, it is also known that the PSD influences the bulk density of a powder collective [31]. The differences in PSD between HR1 and HR4 are comparatively low. Hence, the bulk density difference cannot be influenced by PSD. The WA iron powder with a bulk density of 3.01 g/cm$^3$ ranks between HR1 and HR4. As opposed to HR, the PSD of WA influences the bulk density in addition to the morphology and porosity.

The flow properties of the HR1 iron particles from Figure 9 reached the lowest flow rates with 15.68 s/50 g at the Hall flow measurement. According to the Hall flow test, the flow behavior of the HR4 iron particles is reduced with a measured flow rate of 21.20 s/50 g compared to HR1, despite the more regular particle morphology of the HR4 iron particles. It is generally known that a more regular particle morphology favors a positive effect on the flow properties [1]. Because the PSD and chemical composition between the HR particles types are comparable, it can therefore be assumed that the bulk density has a relevant influence on the flow properties in the Hall flow test. Generally, with increasing bulk density, the vertical stresses of the bulk material in a conical funnel rise, which leads to increased compression of the particles [32]. Due to the higher compression of the particles in the funnel, the interlocking and frictional forces between the particles increase, and as a result, the flow behavior reduces in the Hall flow test. Under the Hall flow test conditions, the influence of the high bulk density on the flow rate is much higher than the influence of a more regular morphology. The flowability of the WA iron particles is the lowest after the Hall flow test with a flow rate of 25.23 s/50 g compared to the HR grades.

In addition to the static Hall flow measurement to determine the flow property, dynamic measurements of an avalanche angle were carried out. The results of the dynamic avalanche angle are summarized in Figure 10.

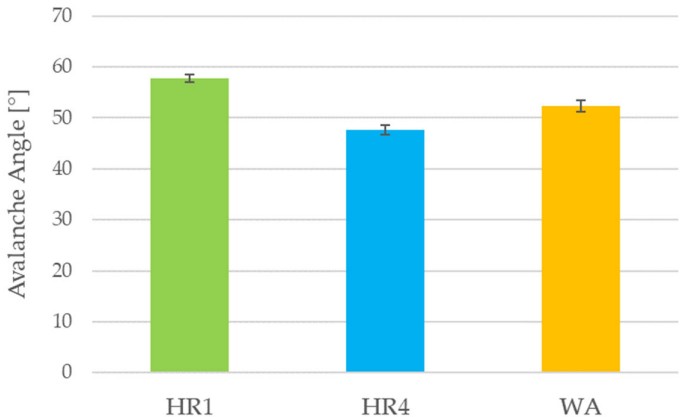

**Figure 10.** Avalanche Angle [°] of the HR1, HR4, and WA iron particles.

With an average avalanche angle of 47.7°, the HR4 iron particles achieves the lowest angle, followed by the WA iron powder with 52.4° and HR1 iron particles with 57.7°. Wouters and Geldart carried out, that an enhanced avalanche angle results in a lowered flowability of the powder [33]. The increased avalanche angle of the HR1 iron particles compared to HR4 is caused by the low bulk density of the particles in addition to the more irregular particle morphology. Macho et al. investigated the influence of the bulk density on the avalanche angle. A comparatively high avalanche angle was observed with a lowered bulk density [34]. The reason for this is the effect of gravity on single particles in the bulk material during an inclined position. Gravity in an inclined position counteracts the adhesive and frictional forces. If the gravitational force exceeds the adhesive and frictional forces, the particles begin to flow. If the gravitational force is increased by denser particles, it can be assumed that the angle at which the particles start to flow decreases [35].

The WA iron powder has larger particles compared to the two HR iron particles, which leads to a lower avalanche angle caused by the number of contact surfaces and thus the potential friction between the particles is lowered [33,36]. However, the influence of the morphology of the WA iron powder on the flow property is more relevant than the particle size. Therefore, the avalanche angle is higher than with the HR4 iron particles. Comparing the results of the flow properties from the Hall flow test (Figure 9) and the RPA test (Figure 10) a contrary flow behavior between the HR iron particle types is visible. On the other hand, the flowability results from the RPA and Hall flow tests for the HR4 and WA iron particles correspond. The difference in the result between the HR1 and HR4 iron particles depending on the measurement method may be caused by the large difference in the bulk density. Within the Hall flow test, the funnel leads to increased vertical stress in the particles, which can have a significant effect on the measurement results of the flow rate in the case of larger differences in the bulk density, so that the comparability of the flowability is not given in this case. Therefore, the RPA is more suitable for determining the flowability in the case of such differences in the properties. Hence, it can be concluded that the HR4 iron particles have higher flow properties than the HR1 and WA iron particles.

Furthermore, the iron particle samples were investigated for compressibility and sintering behavior. The flowability can be derived from the compressibility, similar to the Hausner factor for evaluating flowability. The densities of the loose particle samples, the pressed samples after a pressure of 200 N/mm$^2$ and sintered samples are shown in Figure 11 for the HR1, HR4 and WA iron particle grades.

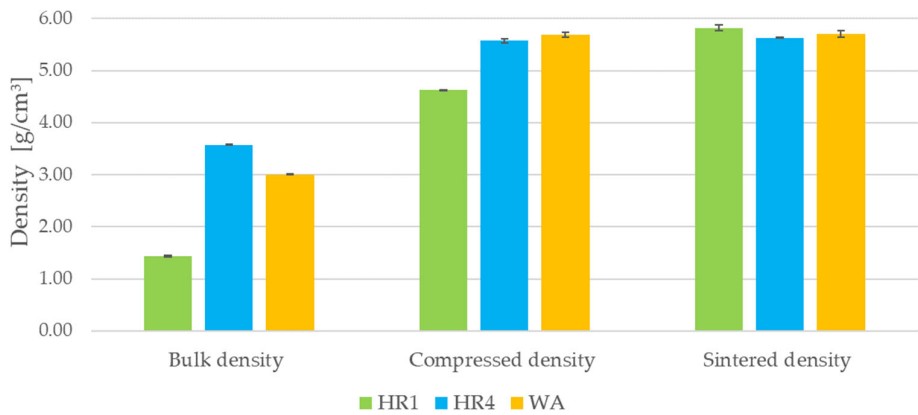

**Figure 11.** Bulk density, pressed density, and sintered density of the HR1, HR4, and WA iron particles.

The increase in density due to pressing is highest for HR1 and lowest for HR4. Thus, the compressibility of the HR4 iron particles is the lowest. It is generally known from the Hausner factor that the lower the compressibility, the higher the flow properties of a bulk material [33]. The results from the determination of the avalanche angle and the estimation of the flow property from the compressibility determination are in agreement. It is noticeable that, for the HR4 and WA iron particles, the density increases from the pressed samples to the sintered samples and is very low compared to HR1. In contrast, the increase in the density of the HR1 iron particles is significantly higher. The density of HR1 increased from 4.62 g/cm$^3$ after pressing to 5.82 g/cm$^3$ through thermal treatment. After sintering, the density of HR1 reaches higher values compared to WA and HR4. It is well known that the specific surface area of iron particles has a significant influence on the shrinkage behavior during the sintering treatment, i.e., the larger the specific surface area, the higher the shrinkage of the pressed particles [1]. The determination of the specific surface by the BET method resulted in a value of 1640 cm$^2$/g for HR1, whereas values of 240 cm$^2$/g and 90 cm$^2$/g were measured for HR4 and WA. Therefore, the high shrinkage behavior of HR1 can be explained by the significantly higher specific surface area of HR1

compared to the HR4 and WA. Concerning this, the high specific particle surface area of HR1 is mainly determined by the open-pored surface structure of the particles.

## 4. Conclusions

In this work, a novel processing route was proposed for manufacturing iron particles using iron oxide powder as a raw material. This iron oxide powder is a by-product of the steel industry, and the proposed process allows for the manufacturing of both spherical dense and irregularly porous iron particle grades. Therefore, this contributes to the circular economy in iron powder production and the conservation of natural resources.

The process allows manufacturing of a variety of iron particle grades through a minor process adaptation in the first preconditioning phase. Two different grades of iron particles, HR1 and HR4, with different properties, were characterized in detail. Regarding flowability, an avalanche angle of 47.7° was determined for HR4 and the bulk density measurements achieved values of 3.58 g/cm$^3$ for this iron particles. Furthermore, the results exhibit spherical morphology, which was analyzed by SEM. The property profile of HR4 results in a high potential for use in additive manufacturing, thermal spray and hot isostatic pressing.

By adjusting the process parameters of the developed process, the irregular iron particles could be manufacturing based on the same iron oxide powder with a comparatively high specific surface of 1640 cm$^2$/g and a low bulk density of 1.23 g/cm$^3$. Those particle properties are suitable for friction applications due to their high specific surface. The specific surface area of typical iron powders for friction applications is less than 230 cm$^2$/g, and achieves to realize the highest possible specific surface area to minimize noise and wear and to increase braking performance [37].

However, more studies and investigations are required to analyze the useability of the presented iron particle grades for the different possible applications. In addition, further development is required through the transfer of laboratory results to the pilot plant and production scale.

**Author Contributions:** Conceptualization, A.W.; Investigation, A.W. and S.P.; Methodology, A.W.; Validation, A.W.; Visualization, A.W. and S.P.; Writing—original draft, A.W., S.P. and S.K.; Writing—review & editing, A.W., S.P., G.W. and S.K. All authors have read and agreed to the published version of the manuscript.

**Funding:** This research received no external funding.

**Institutional Review Board Statement:** Not applicable.

**Informed Consent Statement:** Not applicable.

**Data Availability Statement:** Not applicable.

**Conflicts of Interest:** The authors declare no conflict of interest.

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
