# Peer review of "Manufacturing and Properties of Spherical Iron Particles from a by-Product of the Steel Industry"

_2674-0516, doi:10.3390/powders2020015_

Round 1
Reviewer 1 Report
The manuscript presents a process route used to produce spherical/irregular/porous iron powder from iron oxide powder obtained from hydrochloric acid regeneration. This powder is intended to be used for different technologies and application, such as pressing and sintering, friction, welding, brazing, metallurgical cutting, additive manufacturing.
The manuscript is interesting and well written, but to publish this study first several aspects should be addressed.
1. SI Units should be used in the manuscript.
2. All equipment used for producing or material’s characterization should be presented in the Materials and Methods section, including the method used, name and type of the equipment and producer.
3. Please clarify why was not used the Archimedes method to determine the specimens’ density.
4. Please change the greyscale images to black and white or color images (or you could change the line type or add different markers) because the text is not as visible as it should be and some line are not as visible as they should be.
5. Please add error bars to the Figures where the densities are presented.
6. Please highlight the benefits of the proposed method compared with the existing ones.
7. The authors should consider the possibility to divide the section Results and Discussion in two sections.
8. Please check the manuscript as there are letters missing, units, or capital letters.
Author Response
Firstly, We would like to thank you for reviewing this paper. In the following, we will address your comments and the changes can be viewed in the Word document.
Point 1: SI Units should be used in the manuscript.
Response 1: All units were checked and adjusted with regard to the SI unit. In particular, the units rpm, MPa and °C were changed.
Point 2: All equipment used for producing or material’s characterization should be presented in the Materials and Methods section, including the method used, name and type of the equipment and producer..
Response 2: To improve the clarity of the methodology section, Chapter 2.3 'Equipments for Particle Manufacturing' was added. Previously, the equipment for particle production was described together with the process in the chapter 'Production Method of Iron Powder from Iron Oxide Powder'. In addition, types of equipment and corresponding manufacturers were added.
Point 3: Please clarify why was not used the Archimedes method to determine the specimens’ density.
Response 3: To reduce the investigation effort, the determination of the geometric density was used. A porous surface tends to distort measurements in the Archimedes method due to air entrapment, which makes it necessary to seal the open pores with wax. This leads to an increase in effort. Furthemore, the densities of the samples were determined selectively according to the Archimedes method. No significant differences to the geometric density determination were identified.
Point 4: Please change the greyscale images to black and white or color images (or you could change the line type or add different markers) because the text is not as visible as it should be and some line are not as visible as they should be.
Response 4: The figures have been changed.
Point 5: Please add error bars to the Figures where the densities are presented.
Response 5: Error bars have been added.
Point 6: Please highlight the benefits of the proposed method compared with the existing ones.
Response 6: In the introduction, the benefits were highlighted more strongly.
Point 7: The authors should consider the possibility to divide the section Results and Discussion in two sections.
Response 7: The results build on each other, so that a discussion section between the results contributes to a better understanding in our view.
Point 8: Please check the manuscript as there are letters missing, units, or capital letters.
Response 8: Thank you for the hint. The manuscript was reviewed accordingly.
Reviewer 2 Report
First of all, the work is plausible as new and applied process routes are needed for powder feedstock development. The authors are appreciated for their effort to conduct this research. There are however few fundamental questions emanated from reviewing the paper:
- It is repeatedly mentioned that the methodology used to create an alternative iron-based particle synthesis is less-tedious, cost-effective. However, the reviewer could not find proper justification to back up this claim. For instance, there is still a two-stage process, that involves a number of sub steps. The methodology is very shallow in grasping the whole work, for specific questions please refer to the attached pdf file.
- A formidable miswriting is that the work is attempting to develop a process for particle development, and not powder development (which is the next scale-up work), if successful from the current work. It is a common mistake in literature, whenever fundamental research involves new feedstock development. Therefore, replace the word 'powder' development with 'particle' development or synthesis.
- It will be helpful for the reader to have values and process parameters involved in the stages/sub steps in table format and not hidden texts.
- References are majorly not up-to-date, they are overwhelmingly pre-2010.
Thank you for your good work.

Author Response
Firstly, we would like to thank you for reviewing this paper. In the following, we will address your comments and the changes can be viewed in the Word document
Point 1: It is repeatedly mentioned that the methodology used to create an alternative iron-based particle synthesis is less-tedious, cost-effective. However, the reviewer could not find proper justification to back up this claim. For instance, there is still a two-stage process, that involves a number of sub steps. The methodology is very shallow in grasping the whole work, for specific questions please refer to the attached pdf file.
Response 1: In the introduction, the benefits were highlighted more strongly. According to your suggestions, the process presented in this paper has been deliberately broken down into all of the smallest steps to clarify the influences on powder properties. Additionally, the steps presented focus on the complex reduction step. The methodology section was revised based on the comments in the pdf file.
Point 2: A formidable miswriting is that the work is attempting to develop a process for particle development, and not powder development (which is the next scale-up work), if successful from the current work. It is a common mistake in literature, whenever fundamental research involves new feedstock development. Therefore, replace the word 'powder' development with 'particle' development or synthesis.
Response 2: Powder under development has been replaced by particle.
Point 3: It will be helpful for the reader to have values and process parameters involved in the stages/sub steps in table format and not hidden texts.
Response 3: We added a table format to the process description.
Point 4: References are majorly not up-to-date, they are overwhelmingly pre-2010
Response 4: The older references presents more detailed results. However, we added some latest references.
Reviewer 3 Report
In this study Authors investigated production of spherical iron powder from the by-product of the steel industry. in general the topic is interesting for both researchers and industries. Sustainable manufacturing is an important approach which need to be considered for any manufacturing technology and I believe the presented study is toward achieving this goal. The manuscript is well written with detailed explanation about the results. I have few suggestion for the future studies of the Author:
- it would not be better to consider the same sintering temperature for all the samples. We can see that this parameters affect the results but this is regardless of the quality of the powders.
- It would be good if Authors use the powders for manufacturing a part in even other technologies like AM and compare the result because a challenge toward the sustainable manufacturing is that how different would be the parts quality compared to the one manufactured with ''Fresh powders''
Author Response
Firstly, I would like to thank you for reviewing this paper. In the following, I will address your comments.
Point 1: It would not be better to consider the same sintering temperature for all the samples. We can see that this parameters affect the results but this is regardless of the quality of the powders.
Response 1: The variation in sintering temperature can affect the bulk density of the powder, and depending on the application, a low bulk density (for friction applications) or high bulk density (for additive manufacturing) may be necessary.
Point 2: It would be good if Authors use the powders for manufacturing a part in even other technologies like AM and compare the result because a challenge toward the sustainable manufacturing is that how different would be the parts quality compared to the one manufactured with ''Fresh powders''
Response 2: A similar question is addressed in the context of follow-up activities.
Round 2
Reviewer 1 Report
The manuscript can be published in this revised form.
Reviewer 2 Report
The reviewer is satisfied with the rebuttal and revised manuscript.